# Sustained Complete Response after Biological Downstaging in Patients with Hepatocellular Carcinoma: XXL-Like Prioritization for Liver Transplantation or “Wait and See” Strategy?

**DOI:** 10.3390/cancers13102406

**Published:** 2021-05-17

**Authors:** Alessandro Vitale, Federica Scolari, Alessandra Bertacco, Enrico Gringeri, Francesco D’Amico, Domenico Bassi, Francesco Enrico D’Amico, Paolo Angeli, Patrizia Burra, Quirino Lai, Umberto Cillo

**Affiliations:** 1Hepatobiliary Surgery and Liver Transplantation Unit, Department of Surgical, Oncological, and Gastroenterological Sciences, Padua University Hospital, 35128 Padova, Italy; bertaccoale@gmail.com (A.B.); enrico.gringeri@unipd.it (E.G.); drdamico@hotmail.com (F.D.); domenico.bassi@gmail.com (D.B.); francescoenrico.damico@unipd.it (F.E.D.); cillo@unipd.it (U.C.); 2Unit of Internal Medicine and Hepatology, Padua University Hospital, 35128 Padova, Italy; pangeli@unipd.it; 3Multivisceral Transplant Unit, Department of Surgery, Oncology and Gastroenterology, Padua University Hospital, 35128 Padova, Italy; burra@unipd.it; 4Liver Transplantation Program, Sapienza University, 00185 Rome, Italy; lai.quirino@libero.it

**Keywords:** hepatocellular carcinoma, liver transplantation, downstaging, complete response

## Abstract

**Simple Summary:**

The XXL trial has recently shown that biological downstaging is an effective strategy to also allow liver transplantation into patients with more advanced hepatocellular carcinoma without alternative curative options. Some potential limits of the XXL downstaging protocol are (a) the rather high downstaging failure rate (i.e., 32%), and (b) the additional prioritization of transplantation for patients with a potential good prognosis without transplant, i.e., those obtaining a complete response to downstaging. In this study, we showed that, using aggressive surgical downstaging, it is possible to considerably decrease the downstaging failure rate. Moreover, we showed that it is possible to avoid an immediate prioritization of transplantation for patients with a sustained complete response to downstaging by applying a “wait and see” policy. This policy seems to spare a relevant number of organs without worsening patient outcome.

**Abstract:**

The XXL trial represents the first prospective validation of “biological downstaging” in liver transplantation (LT) for hepatocellular carcinoma. The aim of this study was to compare the Padua downstaging protocol to the XXL protocol in terms of downstaging failure rates and patient outcome. A total of 191 patients undergoing aggressive surgical downstaging and potentially eligible for LT from 2012 to 2018 at our center were retrospectively selected according to XXL trial criteria. Unlike the XXL trial, patients with a complete response to downstaging did not receive any prioritization for LT. Downstaging failure was defined as stable progressive disease or post-treatment mortality. The statistical method of “matching-adjusted indirect comparison” was used to match the study group to the XXL population. Downstaging failure rate was considerably lower in the study group than in the XXL trial (12% vs. 32%, *d* value = |0.683|). The survival curves of our LT group (*n* = 68) overlapped with those of the LT-XXL group (*p* = 0.846). Survival curves of non-LT candidates with a sustained complete response (*n* = 64) were similar to those of transplanted patients (*p* = 0.281). Our study represents a validation of the current Padua and Italian policies of denying rapid prioritization to patients with complete response to downstaging. Such a policy seems to spare organs without worsening patient outcome.

## 1. Introduction

Liver transplantation (LT) represents the best therapy for patients with hepatocellular carcinoma (HCC) regardless of tumor stage [1,2]. Among HCC patients, LT offers the highest survival benefit first to those with decompensated cirrhosis (advanced stage), second to those in the intermediate stage [2,3,4], and more generally to all HCC patients without effective therapeutic alternatives to LT. Patients with early-stage HCC and well-compensated cirrhosis usually represent a low transplant benefit category, since they show great potential for non-transplant curative therapies, such as liver resection or ablation, as alternatives to LT [5]. Intermediate-stage HCC patients have an optimal post-LT outcome when they fulfill validated extended criteria [6,7,8,9] or when they have a good response to a downstaging protocol [10,11,12].

There are two main downstaging strategies. The first strategy aims to bring a patient whose tumor burden is outside accepted criteria for LT to within acceptable criteria (i.e., morphological downstaging). The second strategy aims to select tumors with a good biology and, thus, good outcomes with low risks, irrespective of morphological criteria (i.e., biological downstaging) [10].

While morphological downstaging received strong scientific validation more than 10 years ago by two prospective studies [11,12], until 2020, the scientific evidence promoting biological downstaging remained weak. The XXL trial [13] has great merit in representing the first prospective validation of “biological downstaging” in LT for HCC. This trial has some relevant limits, however. Some limits are strictly ethical, since they concern the use of randomization to deny the best therapy for HCC (i.e., LT) to the unfortunate patients enrolled in the control arm, and others are statistical, since they concern a significantly smaller finally enrolled sample size than the planned sample size.

Other potential limits are strictly clinical and concern the effectiveness of downstaging and LT. First, the XXL trial used almost exclusively nonsurgical loco-regional therapies to obtain a radiological downstaging, while there is now good evidence that mini-invasive surgical therapies (i.e., resection or laparoscopic ablation), when properly selected, offer better results than loco-regional therapies in intermediate-stage patients [14,15,16,17]. Second, in the XXL trial, both patients with a complete (CR) and a partial response (PR) to downstaging procedures were prioritized for LT.

In our center, we have historically adopted an aggressive downstaging protocol [18,19] mainly based on surgical mini-invasive therapies [15,16,17].

Moreover, an alternative approach to XXL biological downstaging, now reaching wide consensus in Italy [19,20], is to give priority of LT only for patients with radiological evidence of HCC after downstaging (PR), while adopting a “wait and see” approach for those with a CR, selecting for LT only CR patients developing HCC recurrence during close follow-up and denying immediate listing for LT to those with sustained CR.

The aim of this study was to compare our aggressive downstaging protocol with a “wait and see” strategy for CR patients with the XXL protocol in terms of downstaging failure rates and patient outcome.

## 2. Materials and Methods

### 2.1. Patient Enrollment

From January 2014 to December 2018, a total of 1390 patients affected by HCC underwent surgical procedures (i.e., liver resection or laparoscopic thermal ablation) at the Hepatobiliary Surgery and Liver Transplantation Unit, University of Padua. This population was retrospectively screened according to XXL trial inclusion/exclusion criteria [13]. As described in Figure 1, almost half of the population was excluded for elderly age, and another relevant proportion was excluded due to Milan criteria at radiology. The final enrolled population, therefore, included 191 patients (Figure 1).

### 2.2. Downstaging Protocol

As previously described, there are two fundamental differences in the management of downstaging and patient selection in our experience compared to the XXL protocol.

The first difference concerns the characteristics of downstaging treatment procedures. In our center, we have, over the years, developed aggressive and multimodal downstaging strategies based mainly on a surgical approach, including in patients with intermediate-stage multifocal HCC [15,16,17,18,19]. In particular, we have given more and more space to laparoscopic microwave thermal ablation, which allows us to aggressively treat single nodules up to 5 cm or more nodules simultaneously with a minimal impact on cirrhosis even in the presence of clinically significant portal hypertension [15,16,17]. As in other experiences [21], liver resection has also been frequently used for downstaging purposes in patients with multifocal HCC, often in association with laparoscopic thermal ablation. Downstaging procedures adopted in the enrolled population of 191 patients are summarized in Table 1. The first treatment our patients usually undergo is surgical (i.e., liver resection and/or laparoscopic ablation), leaving loco-regional therapies only for eventual completion of downstaging when the first treatment was not radical.

As in the XXL protocol, each treatment cycle might include a series of single or combined sessions of surgical or locoregional treatments, and downstaging is considered concluded after multidisciplinary discussion in cases of (1) complete radiological tumor response, (2) best achievable response, or (3) technical infeasibility to proceed. Response to treatments is evaluated at 30 to 90 day intervals by CT scan or MRI, laboratory tests, and measurement of AFP. At the end of the downstaging phase, tumor response is assessed by CT scan or MRI according to the mRECIST criteria. We define “downstaging failures” as all cases of stable or progressive disease (SD and PD, respectively) or patient death within 6 months of the end of downstaging procedures.

The second fundamental difference between our and the XXL downstaging protocols concerns the selection of patients to be placed on the transplant list after downstaging. In the last 15 years, we have given great importance to the response to therapy as a criterion for the selection of patients to be transplant candidates [18,19]. Moreover, after a long and fruitful national consensus that lasted until 2015, all Italian transplantation centers have introduced a selection criterion based on the response to pretransplant therapy; this criterion excludes from transplantation or gives a very low priority to patients with HCC who obtain a complete radiological response after pre-LT therapies [20]. As described in the XXL paper itself [13], this consensus was the main reason for the early termination of the trial.

In our center, in particular, only patients with PR to downstaging (i.e., with the presence of active tumor at radiology) were included in the transplant list, while patients with radiological CR to downstaging were maintained in close follow-up (i.e., without inclusion in the waiting list). The latter patients were placed on the transplant waiting list only in cases of HCC recurrence during follow-up. For this reason, in our center, we also retain the possibility to prognostically evaluate a cohort of patients with sustained CR after successful downstaging who never entered the waiting list for LT, since they never developed HCC recurrence during the study period.

### 2.3. Study Objectives

The primary objective of this study was to assess the effectiveness of our predominantly surgical aggressive downstaging protocol in terms of the proportion of downstaging failures compared to the proportion of XXL protocol failures.

The secondary objective was to evaluate the impact on survival of our peculiar selection protocol (i.e., excluding LT patients with a sustained CR to LT) on patients reaching successful downstaging.

### 2.4. Statistical Analysis

The characteristics of the enrolled population were expressed as medians (range) for continuous variables and as frequencies (%) for categorical variables. The statistical method of “matching-adjusted indirect comparison (MAIC)” [22] was used to make our population similar and comparable to that initially enrolled in the XXL protocol. This statistical method was developed to allow comparison between two studies, controlled for baseline characteristics, when individual patient data are available for only one of the two studies. In this approach, the population that has the individual data available (our study cohort) is adjusted and weighed in such a way that it becomes completely superimposable on the population in which only the aggregated data are available (XXL population). Once the population with individual data is weighed, it is possible to produce comparative evidence by comparing the effectiveness of treatments in the two populations. To achieve this “matching”, we used the “Ebalance” package of the statistical software STATA. For the purpose of this study, we defined the “raw population” as our initial unweighted population, while the “study population” represented the final weighted population.

Because *p*-values can be biased by population size and only aggregate data were available for the XXL population, results from the comparisons between covariate subgroups were reported as the effect size (*d* value); values lower than |0.1| indicated very small differences between means, values between |0.1| and |0.3| indicated small differences, values between |0.3| and |0.5| indicated moderate differences, and values greater than |0.5| indicated considerable differences [23].

For survival analysis, a time horizon from the end date of the downstaging until the death of the patient or last patient’s follow-up (before or after LT) was considered. The date of the last follow-up was 31 December 2020. Follow-up length and survival were expressed as medians (95% confidence interval or interquartile range). The overall survival curves were constructed using the Kaplan–Meier method and compared with the log-rank test. To compare our survival curves with those of the XXL study, the latter were digitized using the software “Engauge digitizer”, and the resulting coordinates were used to reconstruct the individual survival data of XXL patients. This method was developed by Guyot et al. [24].

The statistical significance was set at *p* < 0.05. Statistical calculations were performed using JMP^®^ Pro 15.2.0 (2019 SAS Institute Inc., Cary, NC, USA) and STATA 13.0 (1985–2013 StataCorp LP, College Station, TX, USA).

## 3. Results

### 3.1. Population Characteristics

Upon comparing the general characteristics of our raw population with those of the XXL population [13], the following moderate–considerable differences (*d* value > |0.3|) appeared (Table 2): our population included older patients, a higher proportion of females, hepatitis C virus, and decompensated cirrhosis, HALT-HCC [7] score > 17, and a lower proportion of HCC at first diagnosis just before downstaging.

Using the statistical MAIC method, we obtained a study population (weighted population) from our raw data for comparison with the XXL population. As shown in Table 2, the previous differences between the two populations were no longer evident.

### 3.2. Effectiveness of Surgical Downstaging

Postprocedural morbidity and mortality after downstaging in the study group were very low; the incidence of clinically relevant morbidity (i.e., potentially life-threatening complications or requiring surgical, radiological or endoscopic interventions) was 7.33%, while that of postoperative early mortality (i.e., within 3 months) was 2.09%.

The rate of response to downstaging in the study and XXL groups is shown in Figure 2. In the XXL cohort, the rate of patients with downstaging failure (including 13 tumor progressions during treatment, two patient deaths, and nine patients who dropped out during the observation phase) after downstaging was 32%, considerably higher than that in our study group (12%, *d* value = |0.683|)). The difference between the study and XXL groups in terms of the partial response rate was instead moderate (41% vs. 26%, *d* value = |0.376|).

### 3.3. Survival Analysis

After successful downstaging, 45 XXL patients (61%) were randomized to LT (LT XXL group, *n* = 23) or non-LT (non-LT XXL group, *n* = 22), and they had median follow-up of 71 months (IQR 60–85) [13]. In the study group, 168 patients (88%) had a successful downstaging and they had a median follow-up of 40 months (IQR 34–45).

Among the 168 study group patients, 68 were listed for LT; these included 41 patients with PR to downstaging, and 27 patients with initial CR to downstaging, developing a new HCC recurrence during follow-up. Among listed patients, 51 finally received LT (51/68 = 75%) after a median time from the end of downstaging to LT of 15.0 months (IQR, 6.0–21.4) and eight are still waiting on LT since they obtained a CR after bridging therapies (8/68 = 12%), while nine dropped out from the waiting list due to HCC progression (9/68 = 13%). Among the remaining 100 patients of the study group, 64 patients were not listed for LT due to a “sustained CR” to downstaging (non-LT sustained CR group). The remaining 36 patients were patients with an initial PR to downstaging that, during further evaluation for LT, had a tumor progression contraindicating listing.

Since all 45 tumor progressions (nine dropouts from the WL and 36 initial PR patients with HCC progression contraindicating listing) among 168 successful downstaged study group patients (27%) occurred more than 6 months from the end of downstaging procedures, they were not considered as downstaging failures.

The 1, 2, and 4 year survival rates in the XXL (*n* = 45) and study (*n* = 168) groups were 100%, 71%, and 59% and 82%, 76%, and 56%, respectively (Figure 3a, *p* = 0.818).

We then compared Kaplan–Meier survival curves of the LT XXL group (*n* = 23) to those of the LT study group (*n* = 68), and we did not find any significant difference (Figure 3b, *p* = 0.846).

Moreover, survival figures of the non-LT XXL (*n* = 22) and non-LT Padua (*n* = 100) groups of patients completely overlapped (Figure 3c, *p* = 0.744).

However, when we compared the survival outcomes of the 64 non-LT sustained CR study group patients with those of the 91 LT patients (23 LT XXL + 68 LT study group patients), interestingly, any statistical difference between LT and non-LT groups disappeared (Figure 4a, *p* = 0.281). Conversely, 36 non-LT PR patients had a significantly lower survival perspective than other groups (Figure 4a, *p* = 0.000).

Furthermore, if we considered together our LT group (*n* = 68) and our non-LT sustained CR group (*n* = 64), we obtained a cohort of 132 patients using only 51 organs (51/132 = 39%) with a mid–long-term survival outcome similar to that of the LT XXL-group (Figure 4b, *p* = 0.699) where the organ/patient ratio was 91% (21/23).

## 4. Discussion

The XXL trial represents the first prospective study demonstrating the effectiveness of biological downstaging [13]. This peculiarity of the XXL protocol has enabled its comparison with our population of patients subjected to aggressive surgical downstaging [15,16,17,18,19]. From the analysis of our raw (unweighted) population in comparison with the XXL trial population (Table 2), the following differences emerged:Our population was older (61 vs. 56 years): this can be explained by the fact that only patients with age ≤ 65 years were enrolled in the XXL trial [13], although, in the appendix of the same paper (https://ars.els-cdn.com/content/image/1-s2.0-S1470204520302242-mmc1.pdf (accessed on 13 January 2021)), an upper age limit of 69 years was described as the first inclusion criterion (Appendix A);The rate of HCV cirrhosis patients reported was lower in our cohort, (46.8% vs. 66%). This difference emerged likely because our most recent cohort was probably affected by the progressive decrease in HCV patients due to increasing use of DAA’s eradication therapy [25,26,27];There was a higher number of patients undergoing downstaging for a recurrent HCC (51% vs. 14%). This aspect is probably justified by the fact that ours is a tertiary referral center, and a large part of our case history is made up of patients already undergoing nonsurgical loco-regional therapies (intra-arterial therapy and percutaneous ablation) at other centers for which they are no longer eligible.

To the best of our knowledge, this is the first application of the “matching-adjusted indirect comparison (MAIC)” method [22] in the field of LT for HCC. This method made possible the comparison and a perfect matching between our real population (based on individual data) and the XXL trial population (for which we had only aggregated data).

The first main result of this study was that our aggressive surgical downstaging policy [15,16,17,18,19] resulted in a very low probability of downstaging failures, considerably lower than that of the XXL trial (12% vs. 32%, Figure 2). This result is in line with emerging evidence from the literature proving the effectiveness of surgical approaches also for intermediate HCC [14,15,16,17,18,19,28,29,30]. Although our protocol provides for surgical interventions differently from the XXL trial where the majority of patients underwent loco-regional therapies, our lower risk of downstaging failure was not associated with increased invasiveness (i.e., postoperative morbidity/mortality). This is probably due to the considerable prevalence in our case history of a laparoscopic approach that allows for a lower risk of postoperative decompensation and mortality, as now solidly demonstrated in the literature [15,16,17,18,19,31]. However, it is important to underline that, in the enrolled patients, potentially curative therapies, such as ablation or resection, were not used with a curative intent due to the baseline extension of HCC and underlying liver cirrhosis. In the enrolled patients, conversely, ablation and resection were only used with a downstaging intent in order to decrease the tumor burden and to make eligible for LT patients initially considered beyond transplant criteria.

A second relevant result of this study is that our transplant survival curve completely overlapped with that of the LT-XXL group (Figure 3b). From this perspective, our study represents a sort of external validation of a biological downstaging strategy, as that presented in the XXL trial. This result is also more important considering that we did not give any transplant priority to patients with CR after downstaging. Our peculiar post-downstaging selection protocol was dictated by a center policy in which the response to therapy is a fundamental criterion for the selection of patients to be transplant candidates [19]. The same policy has been recently adopted by all Italian transplant centers [20]. The introduction of this new selection criterion in Italy, in fact, was probably the main reason for the early interruption of the XXL trial [13].

The choice of not giving priority to patients with CR after downstaging is also supported by the excellent prognosis they have in terms of survival without LT; the survival curve of our non-LT sustained CR group (*n* = 64), in fact, was similar to that of LT groups and significantly better than that of 36 non-LT PR patients (Figure 4a).

In a post hoc multivariable Cox analysis of the Mazzaferro’s study [13], the authors estimated that the transplant survival benefit was 26.5 months (95% CI 13.6 to 39.3) in patients presenting with PR and 9.9 months (−5.5 to 25.3) in those presenting with CR after downstaging. In our study, it was not possible to calculate a formal transplant survival benefit in patients with CR after downstaging, since these patients were not prioritized for LT in our experience. However, the absence of a statistically significant difference between the survival curves of the LT group and those of the non-LT sustained CR group (Figure 4a) suggests that, in our study, patients with CR after downstaging also likely have a very low survival transplant benefit.

The most important result of this study in terms of its potential impact in clinical practice is that our policy to give transplant priority to PR patients and to use a “wait and see” approach for CR patients has the potential to spare a relevant number of organs without worsening patient outcome (Figure 4b). The organ/patient ratio was, in fact only 39% in our study vs. 91% in the XXL trial.

On these bases, we strongly believe that our study has at least two clinically relevant significances. First, this study represents a sort of first mono-center validation of the current Italian policy of denying transplant priority to patients with CR after downstaging [20]. Second, our results also potentially have international clinical relevance. In fact, in the majority of international downstaging protocols, especially in the US [11,32,33], patients with CR after downstaging currently have the same prioritization for LT as patients with PR after downstaging. Our study, therefore, has the potential to suggest to other centers worldwide that the best clinical strategy is probably to differentiate the transplant priority of HCC patients after successful downstaging on the basis of radiological response (i.e., PR priority > CR priority).

This study has some important limitations, however. First, this was a mono-center and retrospective study. However, it is important to underline that only three prospective studies on HCC downstaging before LT have been published in the literature, and these studies enrolled 48 [12], 61 [11], and 74 patients [13], respectively. Only Mazzaferro [13] conducted a multicenter study, but it enrolled only 74 patients, where only 23 patients had LT. This evidence suggests that it is very difficult to design and develop a large multicenter prospective study within this very niche topic. Our sample size is considerably larger than that of previous prospective studies [11,12,13], and this aspect makes our results possibly more solid than those of the XXL study, at least from a statistical point of view.

A second limitation of this study is a relatively short follow-up when compared to that of the XXL trial. This relevant difference was due to a different patient enrollment period in the two studies (XXL trial 2011–2015 vs. study group 2014–2018). The reduced follow-up in our population makes the medium–long-term results concerning post-transplant neoplastic relapses less reliable. This limitation is not so relevant, however, considering that there is strong evidence that the majority of post-LT HCC recurrences occur within 24 months from transplant [34,35] and that 24 months was the minimum follow-up of our study group.

Lastly, we had a subgroup of patients with PR after downstaging who were not included in the LT waiting list due to HCC progression during follow-up. This group of 36 patients had a poor prognosis, significantly worse than that of LT or non-LT sustained CR patients (Figure 4a). In the XXL trial, no patients were lost after randomization, probably due to a very short waiting time to LT of 3 months (IQR 2–5) and to the systematic use of sorafenib to stabilize response to downstaging [13]. In our study, median time to LT was significantly longer at 15.0 months (IQR, 6.0–21.4) and we did not routinely use sorafenib after downstaging. It is very difficult to hypothesize the introduction of an XXL-like short waiting time in a real-life clinical situation. However, it is plausible to think that, by giving a higher priority to PR patients and using the new systemic opportunities for HCC treatment [36], the size of this subgroup of patients with a poor prognosis may be reduced in number in the future.

## 5. Conclusions

Aggressive surgical downstaging is effective and safe, with a low probability of failure, in terms of both disease progression and post-procedural mortality and morbidity.

Downstaging based on biological criteria leads to excellent post-transplant survival results, similar to those of patients transplanted at earlier stages of the disease.

Our study represents a validation of the current Italian policy of denying transplant priority to patients with CR after downstaging. Non-LT sustained CR patients have in fact an excellent survival outcome, comparable to that of patients with PR undergoing transplant, and they should be included in the list only in case of tumor recurrence. The main advantage of this policy, avoiding LT to patients with a good prognosis (i.e., sustained CR patients), is to potentially spare a relevant number of organs if compared to the XXL policy.

## Figures and Tables

**Figure 1 cancers-13-02406-f001:**
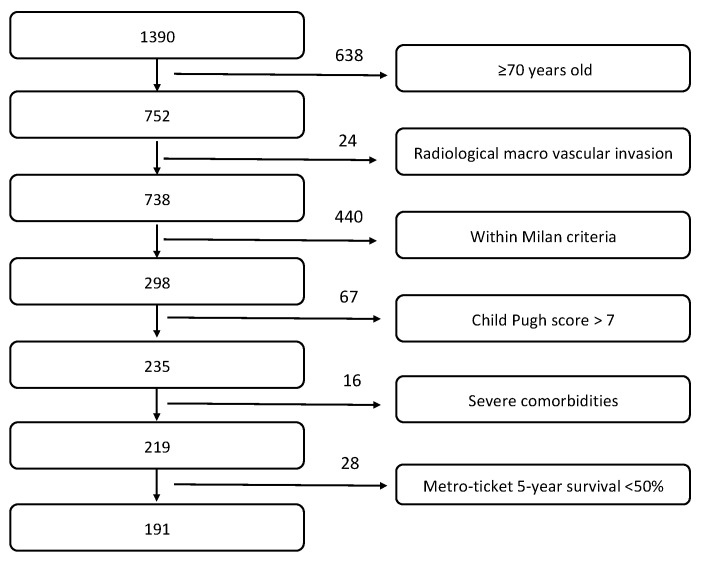
Enrolled population according to XXL inclusion/exclusion criteria [13].

**Figure 2 cancers-13-02406-f002:**
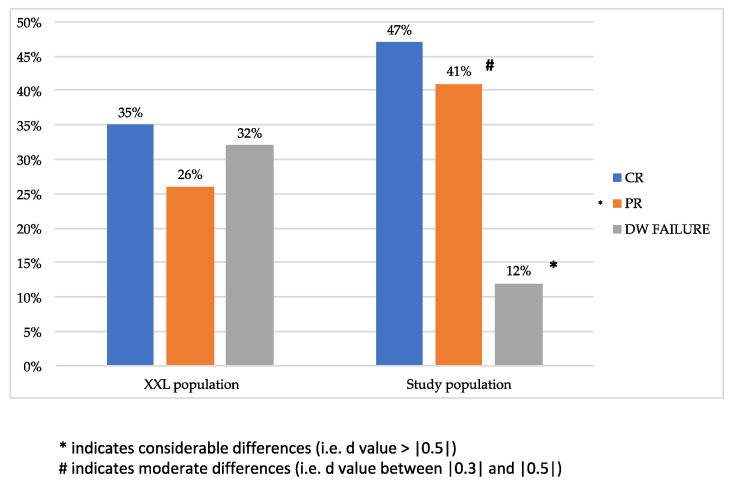
Effectiveness of downstaging in the study and in XXL populations. In five patients of the XXL group (7%), response to downstaging was not described (three developed contraindications to LT, while two withdrew consent) in the study [13]. These patients were not randomized. Abbreviations: CR, complete response; PR, partial response; DW, downstaging.

**Figure 3 cancers-13-02406-f003:**
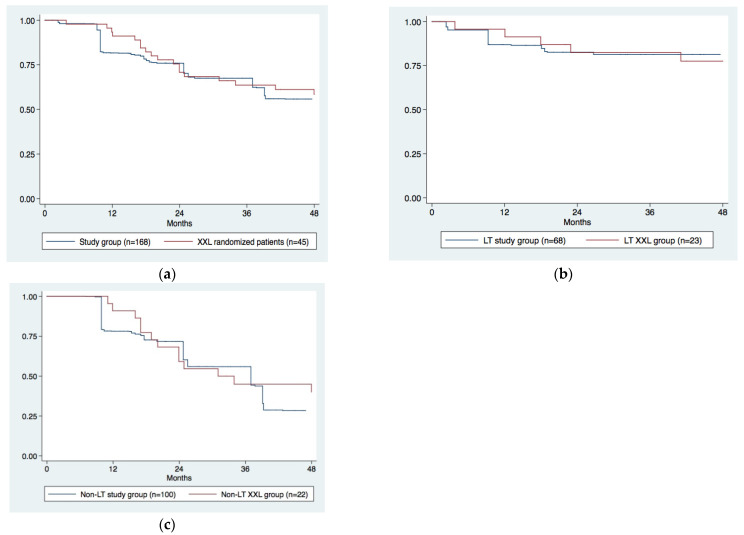
Kaplan–Meier survival curves comparing XXL and study groups: whole population (**a**); patients undergoing LT (**b**); patients in the non-LT XXL and study groups (**c**).

**Figure 4 cancers-13-02406-f004:**
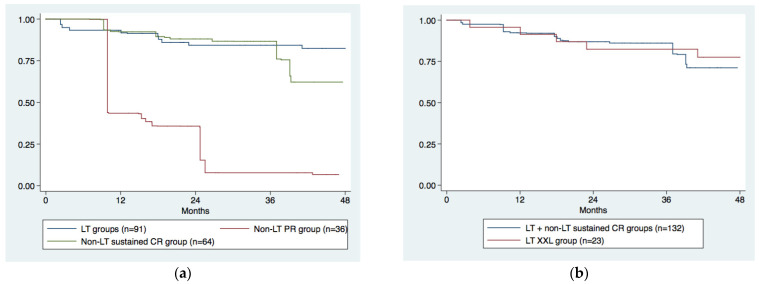
Kaplan–Meier survival curves comparing (**a**) non-LT study sustained CR group (*n* = 64) vs. LT groups (*n* = 91) and (**b**) LT study group + non-LT sustained CR group (*n* = 132) vs. LT XXL group (*n* = 23).

**Table 1 cancers-13-02406-t001:** Summary of downstaging procedures in the Padua population.

Procedure	First Treatment *n* (%)	Other Therapies *n* (%)
Resection ± laparoscopic ablation	71 (37%)	7 (4%)
Laparoscopic ablation	113 (59%)	12 (6%)
Laparotomic ablation	4 (2%)	2 (1%)
Percutaneus ablation	3 (2%)	10 (5%)
Intra-arterial therapies	-	62 (32%)
Total	191	93

**Table 2 cancers-13-02406-t002:** Characteristics of the study group before and after weighting, and comparison with the XXL population.

Characteristics	Raw Population (*n* = 191)	Study Population (*n* = 191)	XXL Population (*n* = 74)
**Age >56 years old**	145 (76%) *	95 (50%)	37 (50%)
**Female sex**	32 (17%) *	13 (7%)	5 (7%)
**BMI**	26 (18–38)	26 (18–38)	26 (19–33)
**HCV**	89 (47%) ^#^	125 (66%)	49 (66%)
**HBV**	41 (22%)	31 (16%)	12 (16%)
**Alcohol/dysmetabolic**	42 (22%)	20 (11%)	11 (15%)
**MELD score**	8 (6–21)	8 (6–21)	8 (6–17)
**Child–Pugh class B**	50 (26%) *	23 (12%)	9 (12%)
**CRPH ***	80 (42%)	95 (50%)	37 (50%)
**First HCC diagnosis**	95 (49%) *	167 (88%)	64 (86%)
**No. of nodules**	3 (1–10)	3 (1–10)	3.5 (1–9)
**Diameter of the largest nodule (mm)**	40 (10–80)	40 (10–80)	41.5 (12–80)
**Sum of diameters (mm)**	70 (26–200)	70 (26–200)	75.5 (13–155)
**Tumor burden**	7 (5–13)	7 (5–13)	7.3 (5.2–13)
**AFP <13 ng/mL**	103 (54%)	95 (50%)	37 (50%)
**BCLC stage B**	158 (83%)	162 (85%)	63 (85%)
**Up-to-7^8^ out**	101 (53%)	104 (55%)	41 (55%)
**UCSF criteria out**	124 (65%)	104 (55%)	41 (55%)
**French Model^9^ high risk (>2 points)**	109 (57%)	97 (51%)	38 (51%)
**HALT-HCC score ≥** **17**	60 (32%) *	1 (1%)	0 (0%)

* indicates considerable differences (i.e., *d* value > |0.5|); ^#^ indicates moderate differences (i.e., *d* value between |0.3| and |0.5|). Abbreviations: BMI, body mass index; HCV, hepatitis C virus; HBV, hepatitis B virus; MELD, Model for End Stage Liver Disease; CRPH, clinically relevant portal hypertension; AFP, alpha-fetoprotein; BCLC, Barcelona Clinic Liver Cancer; UCSF, University of California San Francisco; HALT-HCC, Hazard Associated with Liver Transplantation for Hepatocellular Carcinoma. Portal hypertension is defined as the presence of esophageal varices or platelet counts < 100,000/mL associated with splenomegaly. Tumor burden is calculated as the sum of the number of nodules and the size (in cm) of the major nodule.

## Data Availability

The data presented in this study are available on request from the corresponding author. The data are not publicly available due to privacy restrictions according to Italian law.

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
