# Peer review of "Sustained Complete Response after Biological Downstaging in Patients with Hepatocellular Carcinoma: XXL-Like Prioritization for Liver Transplantation or “Wait and See” Strategy?"

_cancers, 2021, doi:10.3390/cancers13102406_

Round 1
Reviewer 1 Report
Authors aimed to compare the aggressive downstaging protocol with “waiting and see” strategy for CR patients with the XXL protocol in terms of downstaging failure rates and patient outcome. There are several comments to be addressed.
1) XXL protocol seems unfamiliar to readers. It should be described in detail in the manuscript and Table and/or figures.
2) How many patients had the vascular invasion?
3) Considering that patients were initially eligible for curative treatment such as ablation or resection, the clinical significance of the XXL protocol should be interpreted.
Author Response
ANSWER TO REVIEWER 1
Authors aimed to compare the aggressive downstaging protocol with “waiting and see” strategy for CR patients with the XXL protocol in terms of downstaging failure rates and patient outcome. There are several comments to be addressed.
1) XXL protocol seems unfamiliar to readers. It should be described in detail in the manuscript and Table and/or figures.
ANSWER. We thank the reviewer for this comment. As suggest, we have now added more details on the XXL protocol recently published in the Lancet Oncology journal (inclusion and exclusion criteria) in the revised supplementary material.
2) How many patients had the vascular invasion?
ANSWER. None of the patients enrolled in the XXL and Padua studies had vascular invasion. We have now better specified in the revised supplementary material that vascular invasion was an exclusion criterion for both the studies.
3) Considering that patients were initially eligible for curative treatment such as ablation or resection, the clinical significance of the XXL protocol should be interpreted.
ANSWER. We thank the reviewer for this important comment giving us the possibility to better explain our study. In the XXL and Padua protocols only patient beyond Milan criteria were enrolled. In the enrolled patients potentially curative therapies, such as ablation or resection, were not used with a curative intent due to the baseline extension of HCC and underlying liver cirrhosis.
In the enrolled patients, conversely, ablation and resection were only used with a downstaging intent in order to decrease the tumor burden and to make eligible to LT patients initially considered beyond transplant criteria. We added some of these considerations in the revised discussion.

Reviewer 2 Report
- The main concern is the retrospective design and small population for the study. Small population cannot make a clear and solid conclusion. Only 191 patients were enrolled for the study. The exclusion criteria should be described in details.
- Validation study is lacking in the design of the study. Please add the validation study.
- Clinical lab data is lacking. Please provide the information.
- The information of underlying diseases, such as diabetes, hypertension and renal diseases, are lacking. Comorbidity is not analyzed in this study. Further study is suggested.
Author Response
ANSWER TO REVIEWER 2
- The main concern is the retrospective design and small population for the study. Small population cannot make a clear and solid conclusion. Only 191 patients were enrolled for the study.
ANSWER. We understand the reviewer’s perplexities (i.e. single retrospective study). On this field, however, it is important to underline that only three prospective studies on HCC downstaging before LT have been published in the literature, and these studies enrolled respectively 48 [12], 61 [11], and 74 patients [13]. Only the Mazzaferro’s study [13] was multicenter but it enrolled only 74 patients, and only 23 patients had LT. These evidences suggest that it is very difficult to design and develop a large multicenter prospective study on this very niche topic. Our sample size is considerably larger than that of previous prospective studies [11-13], and this aspect makes our results possibly more solid from than that of XXL study at least from a statistical point of view. In the revised discussion we have now integrated the paragraph describing the limits of our study.
- The exclusion criteria should be described in details.
ANSWER. We thank the reviewer for this comment. Our inclusion/criteria were the same of the XXL study. As suggested, in the revised paper we added the XXL inclusion/exclusion criteria in the supplementary material.
- Validation study is lacking in the design of the study. Please add the validation study.
ANSWER. We thank the reviewer for this comment giving us the possibility to better explain the design of our study. We did not make a validation study because our study is itself a validation of other studies. First, it is a validation of the “biological downstaging” strategy proposed in the XXL trial [13]. Second, it was a validation of XXL collateral finding about the transplant survival benefit in CR and PR subgroups. In fact, in a post-hoc multivariable Cox analysis of the Mazzaferro’s study [13], the authors estimated that the transplant survival benefit was 26.5 months (95% CI 13.6 to 39.3) in patients presenting with PR and 9.9 months (–5.5 to 25.3) in those presenting with CR after downstaging. Third, our study represents a sort of first mono-center validation of the current Italian policy of denying transplant priority to patients with CR after downstaging [20]. In the revised discussion we added several sentences explaining better these validation characteristics of our study.
- Clinical lab data is lacking. Please provide the information.
- The information of underlying diseases, such as diabetes, hypertension and renal diseases, are lacking. Comorbidity is not analyzed in this study. Further study is suggested.
ANSWER. We thank the reviewer for these comments. As written above, our study was a validation and, at the same time, a comparative analysis of the XXL trial. On this perspective, we applied the MAIC methodology that is largely used in the oncology literature (one example: Chowdhury S, et al. doi: 10.1007/s12325-019-01156-5.). Also recent papers on HCC have used this methodology to compare the results of real life vs. RCT systemic therapy (Kelley RK, et al. doi: 10.1007/s12325-020-01378-y; Casadei Gardini A, et al. doi: 10.1007/s00432-021-03602-w.). To the best of our knowledge, our study was the first to use of the MAIC methodology in the LT field. Due to the application of this methodology in Table 1 we described only the variables originally described in the XXL trial [3], and specifically those described in the Table S1 of the XXL’s appendix (https://ars.els-cdn.com/content/image/1-s2.0-S1470204520302242-mmc1.pdf). In the XXL trial the only clinical lab described was alpha-fetoprotein level, while comorbidities were not described in detail. It is also important to underline, however, that due to the stringent inclusion/exclusion criteria of the XXL and Padua protocols (see new supplementary material of the revised paper) the impact of liver function parameters and comorbidities on the study population is probably negligible.

Reviewer 3 Report
This study showed the biologic downstaging protocol before liver transplantation of HCC. Authors compared the downstaging failure rates and patient outcomes between Padua-downstaging protocol and XXL-protocol. Downstaging failure rate was considerably lower in the study group than in the XXL-trial (12% vs. 39%). The survival curves of LT-group were similar to that of LT-XXL group. Additionally, survival curves of non-LT candidates with a sustained complete response were similar to that of transplanted patients. Present study validates that current Padua and Italian policies of denying rapid prioritization to patients with complete response to downstaging. I also agreed with such a policy seems to spare organs without worsening patient outcome.
Present study definitely included several limitations such as retrospective single center study and short follow-up period of study patients. However, their study gives transplant priority to PR patients and to “wait and see” CR patients has the potential to spare a relevant number of organs without worsening patient outcome.
Author Response
ANSWER TO REVIEWER 3
This study showed the biologic downstaging protocol before liver transplantation of HCC. Authors compared the downstaging failure rates and patient outcomes between Padua-downstaging protocol and XXL-protocol. Downstaging failure rate was considerably lower in the study group than in the XXL-trial (12% vs. 39%). The survival curves of LT-group were similar to that of LT-XXL group. Additionally, survival curves of non-LT candidates with a sustained complete response were similar to that of transplanted patients. Present study validates that current Padua and Italian policies of denying rapid prioritization to patients with complete response to downstaging. I also agreed with such a policy seems to spare organs without worsening patient outcome.
Present study definitely included several limitations such as retrospective single center study and short follow-up period of study patients. However, their study gives transplant priority to PR patients and to “wait and see” CR patients has the potential to spare a relevant number of organs without worsening patient outcome.
ANSWER. We thank the reviewer for these positive comments. We also understand the reviewer’s perplexities (i.e. single retrospective study with short follow-up). However, as underlined in our study, only 3 prospective studies on HCC downstaging before LT have been published in the literature enrolling respectively 48 (Ravaioli M, et al. DOI: 10.1111/j.1600-6143.2008.02409.x), 61 (Yao F, et al. doi: 10.1002/hep.22412), and 74 patients (Mazzaferro V, et al. DOI: 10.1016/S1470-2045(20)30224-2 ). Only the Mazzaferro’s study was multicenter but it enrolled only 74 patients, and only 23 patients had LT. As you can see, therefore, it is very difficult to design and develop a large multicenter prospective study on this very niche topic. Based on these considerations, as we underlined in the discussion, the limits of our study “are partially balanced by our sample size that is considerably larger than that of the multi-center prospective XXL trial. Our larger sample size makes our results more solid from than that of the XXL study from a statistical point of view. “ Another limit of our study correctly underlined by the reviewer is our relatively short follow-up. As we underlined in the discussion, however, this limit is not so relevant considering that there is a strong evidence that the majority of post LT HCC recurrences occurred within 24 months from transplant [31, 32] and that 24 months was the minimum follow up of our study group. In the revised discussion we have now integrated the paragraph describing the limits of our study.

Reviewer 4 Report
Thank you for having the opportunity to review the manuscript entitled “Sustained complete response after biological downstaging in patients with hepatocellular carcinoma: XXL-like prioritization for liver transplantation or “waiting and see” strategy?”. This is a very interesting analysis from a high-volume hepatobiliary and transplant centre, where an aggressive interventional HCC downstaging approach and LT selection process has been adopted with excellent results that could be helpful for other institutions worldwide. Nevertheless, I would like to point out some concerns:
- If the Authors aim to identify a population from their database which is selected according to XXL Trial criteria, as they state, a cut-off value of 65 years of age should be adopted instead of 70, as reported in Figure 1.
- It is not correct what the Authors state in the first paragraph of the discussion regarding age cut-off values in XXL trial. In Mazzaferro et al manuscript it is stated: “Patients aged 18–65 years presenting with a hepatocellular carcinoma beyond the Milan criteria at nine Italian tertiary care and transplantation centres with availability of all types of therapies for hepatocellular carcinoma were eligible for inclusion.”, in contrast with the Authors’ discussion statement: “the age limit of the XXL-Trial inclusion criteria was 69 years”.
- What do the Authors is the clinical significance of comparing a cohort of patients with both arms of a RCT altogether, where neither of the treatment strategies is the standard in the majority of liver transplant centres and that experienced an early termination due to the agreement of a policy different from both investigated approaches?
- Acronyms in Figure 2 have not be defined, especially “PD” that does not appear anywhere in the manuscript
- The data reported in the Main Text and Figure 2 are questionable and may be misleading. The Authors categorize patients according to the downstaging effectiveness, that appears appropriate when considering their population, but not when considering the XXL Trial results, where a CR+PR rate of 73% (30=240 is reported, and not 51% as the Authors report, since they considered other progressions and dropout at a later stage, resulting in an unfair comparison.
The Authors report as “PD” (progressive disease?) the 29/74 (39%) patients in XXL Trial that were not randomised. This is an unfair comparison and definition, since some of these patients have not been randomised not because of ineffectiveness of the downstaging strategy (for example: 2 withdrew consent without disease progression, 3 developed non-HCC contraindications to transplantation, ...). Moreover, such patients were considered as “PD” only in XXL Trial, since it does not appear that the Authors in their own population categorised as “PD” the 9 patients that had an initial CR/PR and were listed for LT but dropped out because of HCC progression while on the waiting list.
- Was the matching obtained through MAIC validated? If so, how?
- The Authors claim that their approach is linked to a low post-operative/procedure morbidity, but do not present any data in that regard
- I do not think that the main result of this study, as reported by the Authors “..our aggressive surgical downstaging policy resulted in a very low probability of ineffective downstaging, considerably lower than that of XXL trial (12% vs. 39%, Figure 1)” is justified by the results, for the reasons highlighted above.
- It should be acknowledged that the XXL Trial observed a longer post-transplantation gain in survival in patients with PR (26.5 months) than in patients with CR (9.9 months) and that Mazzaferro et al concluded that their findings were supporting the tendency to assign priority to patients with partially responding HCC.
Author Response
ANSWER TO REVIEWER 4
Thank you for having the opportunity to review the manuscript entitled “Sustained complete response after biological downstaging in patients with hepatocellular carcinoma: XXL-like prioritization for liver transplantation or “waiting and see” strategy?”. This is a very interesting analysis from a high-volume hepatobiliary and transplant centre, where an aggressive interventional HCC downstaging approach and LT selection process has been adopted with excellent results that could be helpful for other institutions worldwide.
ANSWER. First of all, we would like to sincerely thank the reviewer for her/his positive comments, but even more for his/her criticisms. Reviewer’s concerns, in fact, gave us the possibility to significantly improve our original manuscript either by clarifying and correcting some relevant aspects.
Nevertheless, I would like to point out some concerns:
- If the Authors aim to identify a population from their database which is selected according to XXL Trial criteria, as they state, a cut-off value of 65 years of age should be adopted instead of 70, as reported in Figure 1.
- It is not correct what the Authors state in the first paragraph of the discussion regarding age cut-off values in XXL trial. In Mazzaferro et al manuscript it is stated: “Patients aged 18–65 years presenting with a hepatocellular carcinoma beyond the Milan criteria at nine Italian tertiary care and transplantation centres with availability of all types of therapies for hepatocellular carcinoma were eligible for inclusion.”, in contrast with the Authors’ discussion statement: “the age limit of the XXL-Trial inclusion criteria was 69 years”.
ANSWER. We really thank the reviewer for this comment giving us the possibility to better explain the controversial point of the age cut-off used in the XXL trial. As underlined by the reviewer, reading the XXL main manuscript (Mazzaferro V, et al. Lancet Oncol 2020. DOI: 10.1016/S1470-2045(20)30224-2) it seems the authors used 65 years old as age cut-off value in their trial. However, the same authors published the trial’s inclusion/exclusion in the paper’s appendix (https://ars.els-cdn.com/content/image/1-s2.0-S1470204520302242-mmc1.pdf) where they wrote as first inclusion criterion: “Cirrhotic patients (any etiology), age 18-69 years, within Child-Pugh class: A-B7”. This discrepancy in the definition of the upper age limit used in the trial was probably due to the fact that during the XXL study period (2011-2015) the upper age limit to accept LT candidates in the waiting list increased in many Italian regions from 65 to 70 years. For these reasons, in our study we decided to maintain “≥ 70 years old” as initial exclusion criterion to select our raw population (Figure 1). From a practical point of view, however, this decision was not so relevant for our analysis, since the MAIC methodology used in our study was able to perfectly match our population to that of the XXL trial also in terms of patients age. As described in table 2, our true “study population” was that obtain after MAIC adjustment, while our initial un-weighted population was defined “raw population”.
In the revised paper, we corrected the first paragraph of the discussion as suggested by the reviewer explaining better the controversial point of the upper age limit in the XXL trial. Moreover, we added more details on the XXL inclusion and exclusion criteria in the revised supplementary material.
- What do the Authors is the clinical significance of comparing a cohort of patients with both arms of a RCT altogether, where neither of the treatment strategies is the standard in the majority of liver transplant centres and that experienced an early termination due to the agreement of a policy different from both investigated approaches?
ANSWER. We understand this reviewer’s perplexity. However, we strongly believe that our study has at least two clinically relevant significances. First, as we already underlined in the first version of our paper, this study represents a sort of first mono-center validation of the current Italian policy of denying transplant priority to patients with CR after downstaging. In other words, our study showed that the political agreement in the Italian transplant community leading to XXL trial early termination was a good “a priori” strategic decision, since this decision has probably spared many organs without worsening patient outcome. Second, our results have potentially also an international clinical relevance. In fact, in the majority of international downstaging protocols, especially in US (11, new ref. Kardashian A, at al. DOI: 10.1002/hep.31210, Mehta N, et al. DOI: 10.1002/hep.30879), patients with CR after downstaging have currently the same prioritization to LT than patients with PR after downstaging. Our study, therefore, has the potential to suggest also to other centers worldwide that the best clinical strategy is probably that to differentiate the transplant priority of HCC patients after successful downstaging based on radiological response (i.e. PR priority > CR priority).
- Acronyms in Figure 2 have not be defined, especially “PD” that does not appear anywhere in the manuscript
ANSWER. As suggested, in the revised paper we have now defined acronyms both in the methods and in figure 2.
- The data reported in the Main Text and Figure 2 are questionable and may be misleading. The Authors categorize patients according to the downstaging effectiveness, that appears appropriate when considering their population, but not when considering the XXL Trial results, where a CR+PR rate of 73% (30=240 is reported, and not 51% as the Authors report, since they considered other progressions and dropout at a later stage, resulting in an unfair comparison. The Authors report as “PD” (progressive disease?) the 29/74 (39%) patients in XXL Trial that were not randomised. This is an unfair comparison and definition, since some of these patients have not been randomised not because of ineffectiveness of the downstaging strategy (for example: 2 withdrew consent without disease progression, 3 developed non-HCC contraindications to transplantation, ...). Moreover, such patients were considered as “PD” only in XXL Trial, since it does not appear that the Authors in their own population categorised as “PD” the 9 patients that had an initial CR/PR and were listed for LT but dropped out because of HCC progression while on the waiting list.
ANSWER. We really thank the reviewer for this specific criticism giving us the possibility to find some mistakes we made in our analysis of downstaging effectiveness and to correct them in the revised paper. As underlined by the reviewer, in Figure 2 we did not actually compare PD between XXL and Padua protocols. Conversely, we compared “downstaging failures”. Also in the first version of our paper in the methods we defined downstaging failure in this way: “.. all cases of stable disease or tumour progression, or patient death within 6 months from the end of downstaging procedures.” So what we described as PD in our original figure 2, it actually included also stable disease and deaths within 6 months after downstaging. We agree with the reviewer that we mistakenly included among downstaging failures also 5 patients (3 developing contraindications to LT, and 2 withdrawing consent).
In the XXL paper (Figure 1, page 951) the authors described 13 tumor progressions during dowsntaging, 2 deaths during downstaging, and 9 dropouts for progression/new lesions during the observational phase of the trial. This specific phase of the trial before randomization was “ a non-intervention period of no less than 3 months” between the end of downstaging and randomization, and the“median duration of the observation period was 3 months (IQR 2.8–3.2)” (page 952, results section). Thus, these 9 dropouts in the XXL trial before randomization were different from our 9 dropouts during the waiting list for LT since our dropouts occurred all more than 6 months from the end of downstaging. Conversely, the early 9 dropouts in the observational phase of the XXL trial, according to our opinion, can be correctly included in our definition of downstaging failure. Based on these considerations we considered downstaging failures of the XXL trial in the revised version of our study 24 patients (13 PD during downstaging+2 deaths+9 dropouts during observational phase). The revised comparison of downstaging effectiveness between XXL and Padua protocols became therefore 24/74 (32%), vs. 23/191 (12%). This confirmed to be a considerable difference (d value = 0.683). We revised the text and the figure 2 according to these new calculations and corrections. We also added this sentence in the results: “Since all 45 tumor progressions (9 drop-outs from the WL, 36 initial PR patients with HCC progression contraindicating listing) among 168 successful downstaged study group patients (27%) occurred more than 6 months from the end of downstaging procedures, they were not considered as downstaging failures.
- Was the matching obtained through MAIC validated? If so, how?
ANSWER. The MAIC methodology is largely used in the oncology literature (one example: Chowdhury S, et al. doi: 10.1007/s12325-019-01156-5.). Also recent papers on HCC have used this methodology to compare the results of real life vs. RCT systemic therapy (Kelley RK, et al. doi: 10.1007/s12325-020-01378-y; Casadei Gardini A, et al. doi: 10.1007/s00432-021-03602-w.). To the best of our knowledge, our study was the first to use of the MAIC methodology in the LT field. As for other matching techniques (i.e. propensity score, or inverse of probability treatment weight) the best method to evaluate the goodness of matching is to measure the effect size/standardized mean difference (D value) between covariates of the study and control groups. D values lower than |0.1| indicate very small differences between means, values between |0.1| and |0.3| indicate small differences, values between |0.3| and |0.5| indicate moderate differences, and values greater than |0.5| indicate considerable differences [23]. This was also the methodology used in our paper and described in the statistical analysis paragraph, in Table 2, and in Figure 2.
- The Authors claim that their approach is linked to a low post-operative/procedure morbidity, but do not present any data in that regard
ANSWER. As suggested, we have now added in the results section the post-operative morbidity and mortality of our aggressive downstaging protocol. Clinically relevant morbidity (i.e. post operative complications requiring surgical, radiological or endoscopic intervention or potentially life threatening) was 7.33%, while post-operative early mortality (i.e. within 3 months) was 2.09%.
- I do not think that the main result of this study, as reported by the Authors “..our aggressive surgical downstaging policy resulted in a very low probability of ineffective downstaging, considerably lower than that of XXL trial (12% vs. 39%, Figure 1)” is justified by the results, for the reasons highlighted above.
ANSWER. As explained above, also in the revised paper we have confirmed that the probability of ineffective downstaging was considerably lower in our cohort than that of XXL trial (12% vs. 32%, revised Figure 2).
- It should be acknowledged that the XXL Trial observed a longer post-transplantation gain in survival in patients with PR (26.5 months) than in patients with CR (9.9 months) and that Mazzaferro et al concluded that their findings were supporting the tendency to assign priority to patients with partially responding HCC.
ANSWER. We sincerely thank the reviewer also for this comment. In the original version of our paper, in fact, we did not elaborate on this interesting collateral finding of the XXL study. We added this paragraph in the revised discussion:
“In a post-hoc multivariable Cox analysis of the Mazzaferro’s study [13], the authors estimated that the transplant survival benefit was 26.5 months (95% CI 13.6 to 39.3) in patients presenting with PR and 9.9 months (–5.5 to 25.3) in those presenting with CR after downstaging. In our study it was not possible a formal transplant survival benefit calculation in patients with CR after downstaging, since these patients were not prioritized for LT in our experience. However, the absence of a statistically significant difference between the survival curves of the LT group vs. the non-LT sustained CR group (Figure 4a) suggests also in our study that patients with CR after downstaging likely have a very low survival transplant benefit. “

Round 2
Reviewer 1 Report
There are major concerns about this article.
Reviewer 2 Report
No further comments.
Reviewer 4 Report
The Authors satisfactorily replied to all the criticisms highlighted and significantly improved their manuscript that would now be, in my opinion, suitable for publication in the present form.